# Bacterial Detection, Deformation, and Torque Loss on Dental Implants with Different Tapered Connections Compared with External Hexagon Connection after Thermomechanical Cycling

**DOI:** 10.3390/ma16113904

**Published:** 2023-05-23

**Authors:** Anselmo Agostinho Simionato, Emerson de Souza Santos, Adriana Cláudia Lapria Faria, Cássio do Nascimento, Renata Cristina Silveira Rodrigues, Ricardo Faria Ribeiro

**Affiliations:** 1Department of Dental Materials and Prosthodontics, School of Dentistry of Ribeirao Preto, University of São Paulo, Ribeirao Preto 14040-904, Brazil; anselmo.simionato@usp.br (A.A.S.); adriclalf@forp.usp.br (A.C.L.F.); cassionasc@forp.usp.br (C.d.N.); renata@forp.usp.br (R.C.S.R.); 2Department of Clinical Analysis, Toxicology, and Food Science, School of Pharmaceutical Sciences of Ribeirao Preto, University of São Paulo, Ribeirao Preto 14040-903, Brazil; emerson.souza.santos@usp.br

**Keywords:** dental implants, X-ray microtomography, dental implant-abutment design, dental abutments

## Abstract

The relationship between bacterial infiltration and internal conical Implant-Abutment Interfaces (IAIs) with different conicities still requires investigations that can offer valuable information in the clinical understanding of peri-implant health. The present study aimed to verify the bacterial infiltration of two internal conical connections with an angulation of 11.5° and 16° with the external hexagonal connection as a comparative after thermomechanical cycling using saliva as a contaminant. Test (n = 10) and control (n = 3) groups were set up. Evaluations were made on torque loss, Scanning Electron Microscopy (SEM), and Micro Computerized Tomography (MicroCT) after performing 2 × 10^6^ mechanical cycles (120 N) and 600 thermal cycles (5°–55° C) with 2 mm lateral displacement. The contents of the IAI were collected for microbiological analysis. There was a difference (*p* < 0.05) in torque loss of the groups tested; groups from the 16° IAI obtained a lower percentage of torque loss. All groups presented contamination and the analysis of the results shows that the microbiological profile of the IAI differs qualitatively from the profile found in the saliva used for contamination. The mechanical loading affects the microbiological profile found in the IAIs (*p* < 0.05). In conclusion, the IAI environment may favor a microbiological profile different from that of saliva and the thermocycling condition may alter the microbial profile found in the IAI.

## 1. Introduction

Osseointegrated dental implants are an excellent option for functional and aesthetic treatment in single, partial, or full dental prosthetic rehabilitation [1,2,3,4,5]. Since the introduction of implants in the market, many implant/abutment connections have been developed to optimize biomechanical performance and preserve peri-implant tissues [6,7]. These connections can be classified as external or internal, depending on the location where the abutment is seated [3]. Internal connections contribute to prosthetic rehabilitations with greater aesthetics, better functioning between components, stability under load, and maintenance of bone height, if compared to external connections, because they have a greater area between the contacting surfaces of the components in the region of the implant-abutment interface (IAI), consequently increasing the frictional resistance [7,8,9].

The success of a prosthetic rehabilitation with dental implants is linked to the maintenance of the peri-implant mucosa and bone tissue. Poor oral hygiene habits, previous periodontitis, and lack of regular maintenance are risk factors for peri-implant tissue inflammation and the development of peri-implantitis and peri-implant mucositis [10,11], which, depending on the host, can lead to deep pockets, bone loss, and treatment failure [11,12]. Biofilms act in the etiology of peri-implant tissue inflammation, as in the periodontal disease. Biofilms with an altered microbiota lead to inflammatory reactions that benefit the microorganisms growth, contributing to further alteration of the microbiota considered normal and increased inflammatory reaction as well, forming a repeating cycle [11]. The consequence over time is that the microbiota of patients with peri-implantitis and peri-implant mucositis differs significantly from that of healthy patients, with a high presence of periodontopathogen bacteria [10,11,12,13]. 

The IAI region has a gap estimated in a range from 0.1 to 10 µm. Considering the bacteria dimensions, the microorganisms passage and installation of biofilms in this region is possible [13,14]. Bacterial colonization in the peri-implant sulcus begins about 30 min after implant installation and establishes a microbiota similar to that found around the natural teeth 2 weeks later [11]. Products of periodontopathogen biofilm metabolism are linked to upregulation of pro-inflammatory and osteolytic mediators [13] in a less vascularized and less resistant mucosa than the mucosa of a natural tooth [15]. The interest in the microbiological behavior of the IAI region resides in the fact that it can serve as a microbiological reservoir that depending on the installed microbiome can increase the concentration of these mediators directly in an area of interest for maintaining peri-implant health and dental implant success over time [5,13,15].

Several connection and abutment designs have been developed, trying to improve biomechanical aspects such as torque loss, screw loosening, stability, and abutment strength [16,17]. The mismatch between the implant parts after the torque loss can cause micrometric movements, generating wear by dynamic friction in contacting surfaces of IAI, leading to fluid flow into the interface by the pumping effect [7,18,19]. Contamination by fluids at the IAI, such as saliva or blood, can decrease the removal torque values of the prosthetic screw, which ends up worsening the pump effect [16,20]. Compared to the widely used external hexagon connections, the internal conical connections were developed to improve the biomechanical behavior, by increasing the contact surface between implant and abutment, improving strength [7].

Currently, the internal conical connections are modified by the manufacturers, mainly in the conicity of the IAI. The taper increase of the internal conical connection increases the contact surface area if the height is maintained [17]; however, it does not necessarily lead to a decrease in torque loss over time, as seen in mathematical models studied [17] and in vitro studies [8,21]. Still, it is not possible to establish relationships between the taper of the internal conical connection and bacterial contamination [22]. Given the lack of in-depth studies on the behavior of internal conical connection designs in biomechanical and microbiological responses simulating applications under stressful conditions, the aim of this study is to investigate the influence of the taper angle of two internal conical prosthetic platforms on the bacterial infiltration into the IAI and deformation comparing to the external hexagon platform control after thermomechanical cycling. The first null hypothesis tested is that there is no difference in the contamination, torque loss, and deformation after thermomechanical cycling between the prosthetic platforms tested. The second null hypothesis tested is that thermomechanical cycling does not produce differences in the testing variables.

## 2. Materials and Methods

The design of this study was carried out considering the evaluation of the influence of the taper of the internal conical connection (11.5° or 16°) using the external hexagon connection as a standard comparison.

Each evaluated group combines one of the connection designs with a different abutment (Table 1).

Dental implants of three different prosthetic connections were used in this study, subdivided into 8 groups (n = 15), where the implants were subjected to thermomechanical cycling (n = 10) or only thermocycling (n = 3), so that the groups could be compared as to the effect of mechanical loading. Implants were reserved to serve as controls for MicroCT (n = 1) and Scanning Electron Microscopy (SEM) analyses (n = 1).

### 2.1. Implant/Abutment Set Preparation

Flowchart of the study is shown in Figure 1. All the implants and abutment components were handled in a laminar flow cabin. The implants were connected to the abutments using manual torque. Then, they were embedded in polyurethane resin using a silicone mold (Silicone Master, Talladium, Curitiba, Brazil) and a parallelometer, with a base perpendicular to a rod attached to a mandrel, ensuring that the implant is embedded with the long axis perpendicular to the polyurethane base. The sets were attached to the vertical rod of the parallelometer mandrel and centered in the silicone mold (Ø25.0 mm × 25.0 mm), which was filled with polyurethane (F160, Axson, Cergy, France). The polyurethane was used due to the similarity of the elasticity modulus of the material with that of the medullary bone [23]. Dental crowns of maxillary canines were fabricated by 3D printing (Sonic 4k, Phrozen, Hsinchu, Taiwan) with provisional crowns resin (PrintaX AA Temp, Odontomega, Ribeirao Preto, Brazil) for each of the abutments tested. All sets had the same final height regardless of connection or abutment. The torque recommended by the manufacturer (Table 1) was performed using a digital torque meter (Torque Meter TQ 8800, Instrutherm, Sao Paulo, Brazil), with confirmation after 10 min. The prosthetic screws were then protected with Teflon tape (Tigre, Rio Claro, Brazil), within 1.0 mm of the upper edge of the abutment. Flow resin was applied up to the edge of the abutment and light-cured for 20 s, filling the prosthetic screw hole. The crowns were fixed using polyether (Impregum™ Soft, 3M, Sumare, Brazil). The use of printed crowns and polyether was established after a pilot test.

### 2.2. Thermomechanical Cycling under Saliva Contamination

Saliva samples (5 mL) from 20 healthy adults (30.55 ± 6.53 years old; 14 females and 6 males) selected from the population of the School of Dentistry of Ribeirao Preto were collected for each group tested. This study was approved by the Research Ethics Committee of the School of Dentistry of Ribeirao Preto—University of Sao Paulo (CAAE: 25836819.2.0000.5419). People who did not present oral manifestations of systemic diseases or acute infectious in the oral cavity were recruited. The saliva was always collected at the same hour of the day, in order to obtain greater uniformity of the saliva samples. After homogenization of the obtained saliva samples, 3.0 mL of saliva was used in each set, enough to cover the implant/abutment interface and resist evaporation during the testing period. Saliva samples for microbiological analysis were collected and homogenized at the beginning and at the end of the experiment.

After embedding the implant/abutment sets and positioning the crowns, transparent acrylic tubes (Ø30.0 mm × 45.0 mm) were fixed with non-toxic acetic silicone (Tytan, Curitiba, Brazil), serving as a reservoir for the human saliva used during the experiment. The sets were positioned in the test container of the Mechanical Cycler for fatigue testing (Biocycle, Biopdi, Sao Carlos, Brazil). 

The specimens were positioned in the machine and the load (120 N) applied to the incisal of the crowns by means of a force applicator with an inclined contact surface (10°), which allows oblique loads additional to those generated by the lateral movement (2 mm) of the test container. Three sets (control) of each type of implant were mounted in the same way as described, but were not subjected to mechanical loading. For 7 days, a total of 2.0 × 10^6^ mechanical loadings were performed at 5 Hz frequency. Approximately 600 thermic cycles (5°–55° C) were performed simultaneously, considering 25 s of filling, 5 min of immersion, and 35 s of draining at each temperature. A total of 3600 mechanical cycles were performed during one thermal cycle. The immersion time of 5 min was used based on a pilot study that investigated the time necessary for saliva to achieve these temperatures considering the temperature transmission through the acrylic tubes used. 

Ten sets of each group were submitted to thermal and mechanical cycling while three sets were only submitted to thermal cycling (control).

### 2.3. Collect for Microbiological Analysis

The sets were removed from the test container and had the crowns removed after abundant washing with distilled water and disinfection by rubbing sterile gauze with ethyl alcohol (70°). The abutments were then disconnected to collect samples for interface contamination analysis using the digital torque meter and the final torque values were obtained. The content inside the implant prosthetic connections and the material present in the prosthetic abutment screw threads were obtained using a microbrush. The tip of the microbrush was sectioned and stored in 0.8 mL TE buffer solution (10 mM Tris-HCl, 1 mM EDTA—pH 8.0) under refrigeration (4 °C).

### 2.4. Torque Loss Evaluation

Before the crowns were positioned, the torques recommended by the manufacturer were performed and confirmed after 10 min (Torque Meter TQ 8800, Instrutherm, Sao Paulo, Brazil). Then, the sets were placed in the test container and thermomechanical cycling was performed. At the end of thermomechanical cycling, the torques required to remove the prosthetic screw (detorque) were recorded. Values of initial (recommended by the manufacturer) and final torques in Ncm were obtained. The results of torque loss of the loaded implant/abutment sets will be expressed as percentage values that indicate how much torque was lost in relation to the initial torque performed, calculated by the following formula:$$T_{L}=\frac{T_{i}-T_{F}}{T_{i}}\times 100$$
where *T_L_* is the percentage of torque loss, *T_i_* is the manufacturer’s recommended torque, *T_F_* is the detorque value.

### 2.5. MicroCT Analysis

The implant/abutment assemblies of the groups were analyzed under microscopy with a stereomicroscope (MSV266, Leica Microsystems AG, Wetzlar, Germany) and light source (CLS 150X, Leica Mycrosystems, Wetzlar, Germany). Images acquired with a camera system (DFC 295, Leica Mycrosystems, Wetzlar, Germany) were used to choose the sets with signs of deformation. A set of tested implants and a control set, composed of new implants and new abutments, were analyzed. The torques applied before the analysis were the detorque values found for each tested set and the values recommended by the manufacturer for the control sets. 

The implant/abutment sets were scanned in the SkyScan microtomograph model 1172 (Bruker-microCT, Bruker Co., Leuven, Belgium). The scanning parameters were as follows: voltage 90 kV, amperage 278 mA, isotropic resolution 8.6 mm, rotation about the vertical axis 180° with a rotation step of 0.5°, total amount of two frames, and use of a 100 µm copper filter. The reconstruction of the axial sections of the angular projections using the modified Feldkamp reconstruction algorithm was performed by the software (NRecon v.1.6.9.18, Bruker Co., Leuven, Belgium) with ring artifact reduction at value 20 (range 0–20), 51% (from 0 to 100%) beam hardening correction, smoothing at value 4 (0–10), and contrast histogram ranging from 0 (minimum value) to 0.20 (maximum value). The images obtained were exported in Windows Bitmap (.bmp) format. Image processing and analysis were performed in DataViewer 1.5.6.2 software (Bruker-microCT, Bruker Co., Leuven, Belgium), which allows simultaneous visualization and correction of the three-dimensional planes (x, y, and z), and export of the images for interpretation.

### 2.6. Scanning Electron Microscopy (SEM) Analysis

One implant/abutment assembly from the control group and one implant/abutment assembly tested from each group were analyzed by SEM. The manufacturer’s recommended torques were applied to the control assemblies and torque values equivalent to the detorque obtained when the tested assemblies were removed from the testing machine at the end of thermomechanical cycling.

The selected assemblies were embedded with medium viscosity epoxy resin (Ohana, Itatiba, Sao Paulo, Brazil) so that they could be longitudinally sectioned in a precision cutter (Accutom-5, Struers, Ballerup, Denmark). The sections were then polished with 600 grain silicon carbide sandpaper (Norton, Saint-Gobain, Guarulhos, Brazil) and cleaned with an ultrasonic bath (Ultrasonic Cleaner 1440D, Odontobras, Sao Paulo, Brazil) for 3 min to remove debris.

For SEM analysis the samples were coated with gold by metallization using a sputtering process (Leica EM SCD050—Sputter Coater, Leica Microsystems, Vienna, Austria) for 120 s in a horizontal position and another 120 s in a vertical position at 0.1 mbar. Then, samples were positioned on the specimen holder table in the scanning electron microscopy equipment (Superscan SSX-550 SEM-EDX, Shimadzu Corporation, Kyoto, Japan) and analyzed according to the lowest possible magnification of the samples determined by the height of the sections obtained (16× to 17×) for a broad view of the region analyzed and at 35× magnification for the areas of interest (prosthetic screw threads and IAI).

### 2.7. Bacterial Detection by 16S rRNA Sequencing

DNA from the samples collected from the implant/abutment interface of each group were combined as DNA libraries for each group. The samples taken from the initial and final saliva of the thermomechanical cycling periods were also combined as two additional libraries. Thus, a total of 18 DNA libraries were processed (8 test, 8 control, initial and final saliva).

The samples collected (containing 0.8 mL TE buffer solution and the bristles of the microbrushes) were vortexed for 30 s (AP 56, Phoenix Luferco, Araraquara, Brazil) and the bristles were removed. The samples were centrifuged for 7 min at 13,200 rpm in a centrifuge (Centrifuge 5427 R, Eppendorf, Sao Paulo, Brazil) to pellet the harvested material and remove the supernatant, agitated for 30 s, and the material after removal of the supernatant was collected. DNA from the libraries was extracted using the CTAB-DNA precipitation method [5]. A new DNA suspension was made with 20 µL of ultrapure H_2_O and 1 µL of RNAse 10 mg/mL. A spectrophotometer (Thermo Scientific Multiskan GO, Thermo Fischer Scientific Oy, Vantaa, Finland) was used to determine the ratio between the absorbances of 260/280 nm.

The V3-V4 regions of the 16S rRNA gene were amplified for each sample by using the polymerase chain reaction (PCR) in a total volume of 25 µL, with universal primers (Integrated DNA Technologies, Coralville, IA, USA): 16S rRNA Forward (TCG TCG GCA GCG TCA GAT GTG TAT AAG AGA CAG CCT ACG GGN GGC WGC AG) and 16S rRNA Reverse (GTC TCG TGG GCT CGG AGA TGT GTA TAA GAG ACA GGA CTA CHV GGG TAT CTA ATC C), Taq polymerase (Fast Start High Fidelity PCR System—Roche) and 20 ng of DNA. Overall, 35 cycles were performed using a thermocycler (Professional Basic Thermocycler, Biometra Ltd., Göttingen, Germany) and respecting the following conditions: denaturation step of 30 s at 94 °C, annealing step of 1 min at 55 °C, and extension step of 1 min at 68 °C. The amplification products (amplicons) were evaluated on agarose gel. The quality of DNA was verified by agarose gel electrophoresis.

The amplicons were purified using A”enco’rt AMPure XP Kit (Beckman Coulter, Pasadena, CA, USA). Libraries were identified using Nextera XT Index kit (Illumina, San Diego, CA, USA) and re-purified by Agencourt AMPure XP Kit for the Real-Time PCR technique and were quantified by real-time qPCR—NEBNext Library Quant Kit for Illumina (New England Biolabs, Ipswich, MA, USA) and fluorimetry using Qubit 2.0 (Thermo Fischer Scientific Oy, Vantaa, Finland) to determine library concentrations. The DNA sequencing (2 × 251 bp) of the samples was performed in the MiSeq Sequencing device (Illumina, San Diego, CA, USA). Sequencing was processed and interpreted using QIIME2-2022.2 software [24]. Low-quality sequences, short fragments, chimeras, and polyclonal sequencing were excluded. After sequence alignment, the libraries were grouped into phylotypes using the Sklearn-based taxonomy classifier method in the Silva 138 SSU [25] database at 99% identify for taxonomy assignment. The mean length of the reads obtained was 438 bp, the minimum 345 bp, and the maximum 447 bp.

### 2.8. Statistical Analysis

Statistical analysis of the torque loss data was performed by IBM SPSS Statistics (SPSS v20.0, IBM) and presented a normal distribution and homogeneity of variances. The statistical test applied was the One-Way ANOVA with Tukey post-test. The 16S sequencing data obtained of the libraries were analyzed with QIIME2-2022.2 and differential abundance with Linear discriminant analysis Effect Size (LEfSe) [26]. Kruskal–Wallis test based on Shannon index and PERMANOVA based on unweighted Unifrac were used to check the diversity of the analyzed libraries. The images obtained by SEM and MicroCT were interpreted qualitatively.

## 3. Results

The results of the torque loss analysis after thermomechanical cycling are presented in Figure 2 and Table 2. The results refer to the average percentage of torque loss in relation to the initial torque.

Statistical analysis showed that there is a significant difference between the groups tested (*p* < 0.05). Group G2 showed difference with groups G4 (*p* = 0.008), G5 (*p* = 0.012), and G6 (*p* < 0.05) and group G3 showed difference with groups G4 (*p* = 0.018), G5 (*p* = 0.027), and G6 (*p* < 0.05), where G2 and G3 presented lower percentage of torque loss in the prosthetic screw after thermomechanical cycling. Group G6 showed a difference (*p* = 0.013) with group G8, showing a higher percentage of torque loss after thermomechanical cycling.

Evaluation by 16S rRNA gene-based metagenome approach showed that there was bacterial content in all the libraries evaluated. A total of 18 phyla, 22 classes, 62 orders, 112 families, and 234 genera were identified in the analyzed libraries. A total of 915,253 sequence reads were obtained from the processed library set, with a mean of 50,847 sequence reads per library analyzed. Shannon’s diversity index, which describes the alpha diversity considering the number of genera and abundance (Figure 3), showed that there was no difference (*p* = 0.616) between the diversity of the libraries collected from the IAI, regardless of mechanical loading or prosthetic platform. All the libraries presented a high diversity of microorganisms, with Shannon index values ranging from 4.28 to 6.09, with a mean of 5.27.

Figure 4 and Figure 5 shows the relative abundance of the phyla and genera found in the libraries collected from the different IAI’s analyzed. Both figures also have the relative percentages of the initial and final saliva libraries as a qualitative reference. The phylum *Proteobacteria* presented the highest relative abundance (86.42%) in libraries obtained from the IAI’s submitted to thermomechanical cycling. The libraries obtained from the initial and final saliva showed a considerable relative abundance of the *Firmicutes* (56.50%), *Bacteroidota* (20.08%), and *Proteobacteria* (8.05%) phyla.

Acinetobacter (14.84%), Bacillus (0.64%), Comamonas (3.12%), Delftia (0.42%), Domibacillus (0.79%), Enhydrobacter (24.49%), Massilia (2.01%), Neisseria (1.83%), Paracoccus (0.21%), Pelomonas (0.49%), Pseudomonas (24.85%), Ralstonia (2.23%), Sphingobacterium (1.50%), Sphingobium (0.56%), Stenotrophomonas (4.98%), and Streptococcus (2.49%) showed more relative abundance of genera in libraries obtained from IAI’s.

The PERMANOVA statistics show that unweighted Unifrac analyses, a beta-diversity measure that uses phylogenetic information to calculate the distance in structure between compared biological communities, indicated differences (Pseudo-F = 1.28, *p* = 0.042) between the treatment groups (mechanical loading) in terms of bacterial community structure, however no specific differences (*p* > 0.05) were shown in the pairwise comparison between the analyzed libraries. The PCoA graphical representation of the Unweighted Unifrac analysis results (Figure 6) shows that the mechanically loaded implant groups showed a different area of concentration than the control groups and that the initial and final saliva groups are distant from the other groups and close to each other, which indicates that the bacterial profile found in the IAI differs from the saliva used for contamination. Groups G6, G7, and G8 without mechanical loading show their profiles distanced from the area of higher concentration, approaching the saliva samples.

A discriminative rate between groups based on relative abundance was obtained by analysis with LEfSe (Figure 7). Comparisons (*p* < 0.05, LDA > 2.0) were made between the libraries analyzed, treatment (mechanical loading), and the prosthetic platform used. There was no difference (*p* > 0.05) when comparing the different groups of implant/abutment test and control sets. Mechanically loaded dental implants showed a higher relative abundance (*p* < 0.05) of the genera *Comamonas* and the controls had a lower relative abundance (*p* < 0.05) of the genera *Lawsonella*, *Massilia*, and *Ralstonia*. Dental implants with External Hexagon (EH) prosthetic platform showed higher relative (*p* < 0.05) abundance of *Achromobacter*, *Selenomonas*, *Enterococcus,* and *Flavobacterium*. Implants with a 16° IAI had a lower relative abundance (*p* < 0.05) of *Allorhizobium-Neorzobium-Pararhizobium-Rhizobium* and *Sphingobacterium*.

MicroCT and SEM analyzes showed that no microgaps were seen in the IAI region and in the contact region between the screw head and the abutment, as can be seen in Figure 8 and Figure 9.

The MicroCT and SEM analysis demonstrated the existence of adaptation of the abutments to the prosthetic platform in the control groups even after thermomechanical cycling, without deformation in the region of the IAI and the morphology of the prosthetic screws used for each platform are different. The groups with 11.5° IAI (CM—G4, G5 and G6) show shorter screws with less thread contact compared to the other platforms tested (Figure 10).

Qualitative analysis of the images obtained through on the dental implant used as a control and on the dental implant after thermomechanical cycling showed that there is a void space in the IAI region on G2 control and G2 test groups. This is possibly due to the manufacturing process or abutment design (Figure 11).

## 4. Discussion

The angulation of the internal conical connection between implant and abutment can influence the biomechanical behavior and contributes to factors related to the maintenance of the dental implant, prosthesis, and peri-implant tissues. The torque loss of a prosthetic screw in implant-supported prostheses is related to some mechanical and biological problems, which are connected, as in peri-implant disease. This problem is linked to the amount of bacterial invasion IAI, especially in the external hexagonal connection. This study observed that thermomechanical cycling affected bacterial infiltration of the IAI in the three prosthetic platforms tested. Some genera of bacteria were more abundant in the groups subjected to thermomechanical cycling and in the external hexagon (EH) platform while some genera were less abundant in the groups without mechanical loading and with an internal conical design with a 16° IAI (GM). It was also possible to identify by SEM and MicroCT analyses a void space in one of the groups of the GM connection. In this context, both the null hypotheses tested were rejected.

The conical connection can bring clinical benefits derived from the increase in static friction between the components, preventing the torque loss of the prosthetic screw and fractures, which maintains stability during the action of masticatory stresses even with torque decrease over time [8]. The coefficients of friction between the surfaces of the internal conical platform, which can be related to the mechanical properties of the materials used in the components, area and roughness on the contact surface, external contaminants and the recommended torque, directly affecting the detorque values [17,19,21], where the area of contact formed between the prosthetic platform of the implant and abutment seems to have an effect on the connection efficiency [9,21]. Two internal conical connections with an angulation of 11.5° and 16° were used and lower values of torque loss were found in two groups of implants with a 16° IAI (G2 and G3), showing a difference with all other groups of implants with an 11.5° IAI. These results are possibly not linked to the taper angle of the internal conical prosthetic platform, but to other factors that differ between the two implants, such as the recommended torque and the contact area between the screws threads, regardless of the taper angle. The prosthetic screws threads used with the 11.5° IAI are smaller than those used in the 16° IAI, which decreases the contact area and possibly contributed to the results found.

The variation in taper angulation of internal conical connection does not seem to alter the bacterial count and frequency in the IAI after thermomechanical cycling [22]. Apparently, it is much more a matter of quality in the manufacture of the components, use of compatible components, and prosthesis obtention processes [27]. In the present study, no inferences can be made between the angulation of the internal conical connections and the amount of infiltration that may have occurred, due to the methodology used in processing the samples. However, when comparing the microbiological profiles found in the IAI of these connections, it is not possible to say that they do not have differences. A limitation of this study regarding the biological analyzes was the number of libraries generated for each group. It is possible that, if they were more numerous, more evident differences could be found among the different groups and platforms. It is important to point out that G2 and G8 groups use components that will be submitted to laboratory waxing and casting procedures, knowingly increasing the potential of gaps and mismatches [25], which was not performed in the methodology of this study.

The present study performs a lateral movement that provides forces in a non-axial direction during the test. In recent studies [7,28,29], the force applied on implants during mechanical testing varied from 50 N to 200 N, the frequency from 1 Hz to 2 Hz, and the number of loading cycles was from 150,000 to 500,000 under static or dynamic conditions. Previous studies [28,30] stipulate that 150,000 mechanical cycles being performed at 1.6 Hz under a static load of 50 N and 45° are equivalent to 1 year of masticatory forces in the anterior dentition. The methodology employed in this study performs a substantial number of cycles and high loading frequency. This may contribute to the results of torque loss, and possibly intensified pumping effect, even without plastic deformations in the prosthetic platform. If there was a large enough opening during load application and saliva passage, this may have contributed to the torque loss [16,20]. The present study uses only 2-piece different abutments for each prosthetic platform, however, some components use a taper integrated screw-in prosthetic abutment, known as single-body abutments, offering resistance to torque loss in single-unit restorations [17].

The microgap formed between the abutment and the implant independently from the prosthetic connection can serve as a reservoir for biofilm [1,15] and as a complicating factor during implant function under loads [14]. Factors such as implant material quality and selection, imprecision in technical design or manufacturing, or association between dental implants and non-original abutments can contribute to the existence of microgaps [27]. It is documented that periodontopathogen bacteria are about 0.2 to 1.5 µm wide and 1 to 10 µm long [15,31]. Under free loading conditions, the degree of contamination is totally dependent on the adaptation between the components [18]. Prior to any loading application, it is documented that the microgap between implant and abutment is 0.1 to 10 µm, increasing over time as loading is applied [13,14]. The other groups, as can be seen in the MicroCT and SEM analyses, did not show any deformation. The preparation of the samples for the two analyses was performed under static conditions and with the torque values found in the final condition of the experiment in the test groups, or recommended by the manufacturer in the control groups. Even so, this study used a vertical load application methodology and lateral displacement, that may have caused abutment displacement and the opening of microgaps, even briefly, until the implant returns to its initial position, which may allow or increase the degree of bacterial microleakage in the region, besides causing friction between the two components on the opposite side, rotational movements on the abutment, and torque loss in the prosthetic screw [3,14]. Comparing the size of the previously found microgaps and the size of the bacteria, contamination was expected, even in load-free conditions, as already detected [32,33].

The bacterial profiles found in the libraries of the analyzed IAIs show differences from the libraries obtained from the saliva samples used for contamination during the testing period. Possibly, the environment created by closing the prosthetic interface of the implant with the abutment favors the survival and colonization of some bacterial genera by decreasing the flow of oxygen and nutrients. The bacterial genera detected in larger percentages show cytotoxin production capacity intensified under the conditions created in the IAI. Three bacteria genera showed a high percentage of presence (64.18%) in the analyzed libraries. The results highlight the presence of the *Pseudomonas* genus in the analyzed IAI libraries, which is also found in saliva for contamination after the testing period (saliva final group). Deep periodontal pockets and peri-implantitis sites are usually dominated by Gram-negative anaerobic bacteria [11]. *Pseudomonas* genus shows growth in anaerobic environments [34] and is linked to infections such as peri-implantitis, mainly by the *Pseudomonas aeruginosa* species, which is Gram-negative motile bacteria and is able to produce inflammatory responses and inflammation-related cytotoxins [2,35,36]. *P. aeruginosa* shows an interspecies commensal relationship in biofilm formation with *Streptococcus* genus, also found in the analyzed libraries, which limits the virulence of *P. aeruginosa*, but improves disease conditions such as lung cystic fibrosis [37]. *Acinetobacter baumannii*, a species moderately found in the analyzed libraries, was found together with *P. aeruginosa* in association with aggressive forms of periodontal disease, especially when these bacteria were found associated with the red complex bacteria [36], and is associated with biofilm formation and stimulation of host immune response [38]. *A. baumannii* is preferentially aerobic; however, when encountering hypoxic conditions, it shows the production of virulence factors that can intensify an inflammatory condition [38]. The *Enhydrobacter* genus has only one known species, *Enhydrobacter aerosaccus*, and was detected in the analyzed implant libraries. It is a gram-negative facultative anaerobic species found in the epithelial microbiota [39].

Analysis of the relative abundance of bacterial genera present in the processed libraries shows that the bacterial profile found in the IAI seems to be different from that found in the saliva that was used as a contaminant. The profile found for the loaded and control G8 groups apparently also differs from the other implant groups, with the profile of the control G8 being more similar to that of the saliva used for contamination, which could indicate greater saliva leakage into the IAI in the absence of load during the testing period. The G8 group consists of the set of implants with external hex connection combined with UCLA abutments. It is found in the literature [32,40] that the external hex connection has larger gaps and greater contamination in the IAI region, which may explain the results found.

The maintenance of implants over time is dependent on the preservation of the crestal bone level, which shows better results when dental implants with internal IAI are applied [41,42], especially those with an internal conical connection [41]. A more diverse microbial profile in analyses of internal implant contents is related to healthy implants compared to implants with peri-implantitis [13]. Peri-implant biofilms exhibited significantly lower diversity than healthy and periodontitis subgingival biofilms [12]. The results obtained show that the groups with external hexagonal connection show microbiological profiles more similar to those found in saliva used for contamination than those with internal conical connection, especially in G8, which shows the most similarity between the bacterial profiles taken from the IAI and those obtained from the saliva sample analysis, that in a superficial analysis would indicate that it could present a greater bacteriological diversity. However, there was no more diversity in these samples, because there was no significant difference between the diversities of the libraries analyzed. Further, it is necessary to consider that the microbiota was developed during thermomechanical cycling (7 days), a relatively short period compared to the useful life of a dental implant. The period corresponding to 7 days can be related to a postoperative stage and may be a useful indicator of the early stages of microbiota evolution in the IAI. There are many more factors to consider in the development of the microbiota and in the execution of this in vitro study, which relied on a few libraries from each group, a limited number of control implants, and a methodology adapted and restricted to the laboratory conditions, thus not reflecting the complexity and dynamics of the conditions that these materials encounter when applied.

The results of the unweighted Unifrac and LEfSe analyses demonstrate that mechanical loading influenced the bacterial dynamics in the tested assemblies. It is possible to see in Figure 6 that the distribution of the control groups is different from the groups submitted to loading, in addition to the relative abundance of some generics differing between the groups. *Comamonas* is a genus previously found in increased amounts in the subgingival microbiota of patients with a history of periodontitis compared to healthy patients [4]. The mechanically loaded implant group showed higher relative abundance of *Comamonas*. Even without statistical differences, it was observed that the G2 test group showed a higher relative abundance of this genus and was possibly determinant for the difference found between test and control groups. From the MicroCT and SEM analyses, it is possible to observe the existence of a void space in the adaptation between implant and abutment in G2, even in the control groups, which may have potential to provide favorable conditions for this genus during the cycling period. Another important aspect that may indicate the influence of mechanical loading on bacterial leakage into the IAI is the lower presence of the genera *Lawsonella*, *Massilia*, and *Ralstonia* in the control groups. It is reported that genera *Lawsonella* and *Massilia* can be found in relatively higher abundance in supragingival biofilms from patients with peri-implantitis [10]. The results obtained during the test period in this study show that mechanical loading produced bacterial invasion of peri-implantitis-related species, probably due to factors mentioned above, as the pump effect, and by the presence of pre-existing factors, as the void space found in G2.

The LEfSe analysis between the platforms showed that GM groups shows lower relative abundance of *Allorhizobium-Neorzobium-Pararhizobium-Rhizobium* and *Sphingobacterium*, whereas EH shows higher abundance of *Achromobacter*, *Selenomonas*, *Enterococcus*, and *Flavobacterium*, of which *Enterococcus* [10,13] and *Selenomonas* [10] were found in increased relative abundance in the supragingival biofilm of peri-implantitis. However, *Selenomonas* are also found in larger quantities in the biofilm of patients with gingival health when compared to biofilms from patients with peri-implantitis and periodontitis [4,12,13]. *Enterococcus* can condition the environment for other species and shows ability to form biofilm in vitro by producing oxygen and lactic acid [13]. Considering the test period of this study, it is possible to infer that the microbiological profile of the EH groups (G7 and G8) could be healthier due to the similarity to the microbiological profile found in saliva; however, with the development of this microbiota over a longer period of time, it is possible that it will be able to cause inflammatory reactions in the peri-implant tissue. It should be noted that the development of peri-implantitis is multifactorial and individual-dependent, and the mere presence of some bacterial genera does not determine the disease because the development of a microbiota is a complex process. The present study only points to bacterial genera with significant differences in relative abundance in the results found under specific experimental conditions, but in future studies, they could indicate bacterial genera to be used as targets in more specific methodologies. GM showed a lower relative abundance of two bacterial genera while EH showed a higher relative abundance of four genera and 11.5° IAI showed no difference with the others and, given the methodology used, it is not possible to say that the 16° IAI is more efficient in sealing the IAI than the other connections tested.

In prosthodontics on implants using abutments, two interfaces are created: IAI and the interface between abutment and prosthesis. There are single-body implants that allow for the creation of only the interface between implants and prosthesis, which would not allow for bacterial growth at the IAI, which does not exist [43,44]. This is an advantage in the peri-implant health of the rehabilitation performed, since it reduces the sites where bacterial invasion would occur and problems such abutment screw loosening are non-existent. At the same time, short implants combined with angled abutment are applied to avoid important anatomical areas [44]. The present study used only the conventional implant/abutment system with a conventional height, and future studies may be interesting to compare single-body and IAI implants in the biological and biomechanical related aspects, as well as studies in the same direction with short implants and angled abutments, including the possible treatment combinations. Clinically these are interesting and successfully solutions to issues presented in this study.

The present study attempted to simulate the oral conditions by considering the dynamics of varying temperatures and the application of oblique loads simulating forces produced during masticatory efforts. Furthermore, saliva was used for contamination rather than specific microorganisms, which may help in understanding the dynamics of bacterial infiltration in IAI. Even with all these methodological precautions, the conditions of the oral environment are still unique and present difficulties to be perfectly replicated during an experiment. Food habits, salivary flow, and hygiene habits conditions that could alter the concentration and provide some nutrition to the microorganisms in the oral environment were not present in this study during the thermomechanical cycling period. The use of temporary crowns in the methodology applied in this study is a limitation of this study, once tue use of permanent crowns and luting agents could change the transmission of applied forces to the system and thus there could be different results for bacterial profiles. While the possibility of using a definitive material is justified by collaborating in the understanding of bacterial development, the present study, after pilot test, opted for provisional crowns fixed by polyether because of the ease of removing the crowns after thermomechanical cycling without procedures that could damage the IAI by vibration, impact, or creation of levers on the abutment.

Despite these limitations, the results were important for understanding the bacterial profile over a 7-day period in three different connections and showed that the angulation of the internal conical prosthetic connection apparently were not determinant for development of the bacterial profile, although the connection with greater angulation showed less relative abundance of two bacterial genera. Clinically, the results show that regardless of the prosthetic connection there is bacterial infiltration into the IAI and that this must be considered in the development of peri-implant diseases and the maintenance of a prosthesis on dental implants. Further studies would need to be performed to test the angulation in the prosthetic connection, using quantitative methods of microbial counting, which could be more elucidative regarding the stability of the connection and the ability to reduce bacterial flow in the IAI. Indications that the bacterial profile found in IAIs is different from that found in saliva were shown in the results of this study, even using saliva from healthy participants. This is another point that can be explored in future studies.

## 5. Conclusions

In view of the limitations, the present study suggests the following:The external hexagonal connection and the internal conical connections tested, regardless of IAI angulation, allow bacterial infiltration;The bacterial profile found in IAIs differs qualitatively from that found in saliva, with the presence of bacterial generics linked to peri-implantitis;Thermomechanical cycling as performed modifies the bacterial community structure found and does not lead to deformation in the IAI;The environment created by the IAI may favor the growth and survival of some bacterial genera.

## Figures and Tables

**Figure 1 materials-16-03904-f001:**
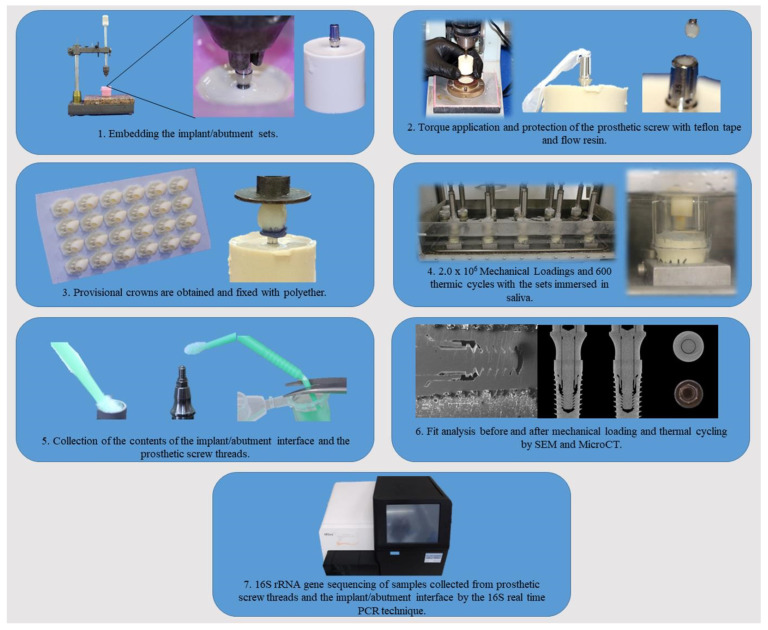
Flowchart showing the steps performed in the experiment.

**Figure 2 materials-16-03904-f002:**
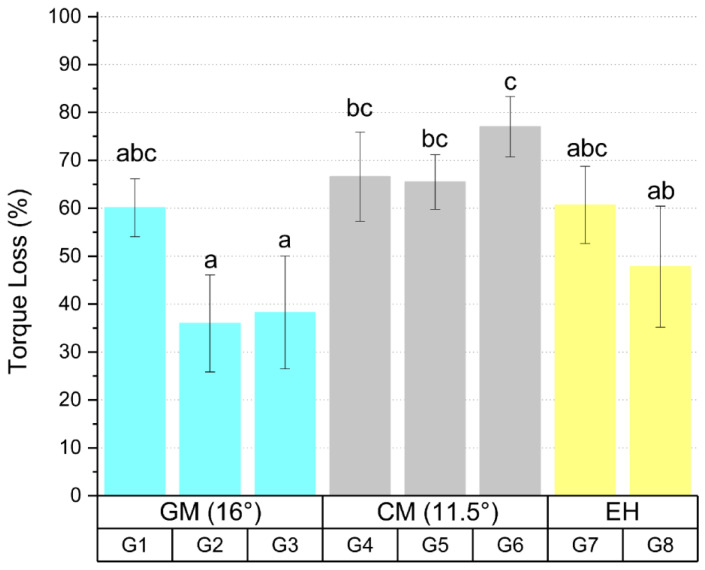
Torque loss mean values and standard deviation of the tested groups after thermomechanical cycling. Different letters indicate statistical difference between the groups.

**Figure 3 materials-16-03904-f003:**
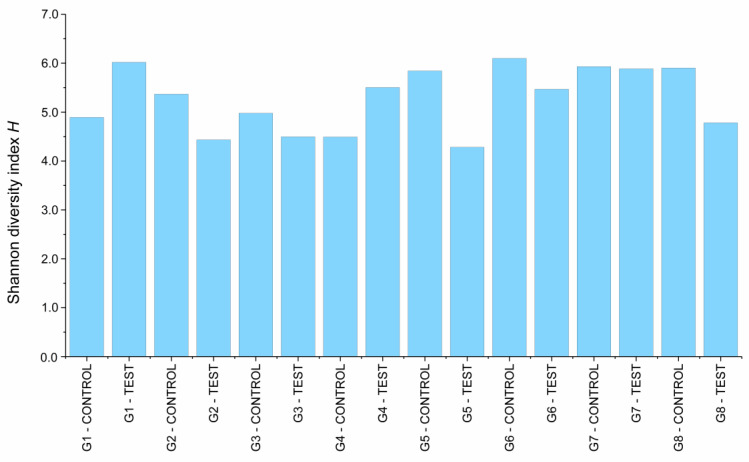
Shannon’s diversity index calculated for the implant libraries.

**Figure 4 materials-16-03904-f004:**
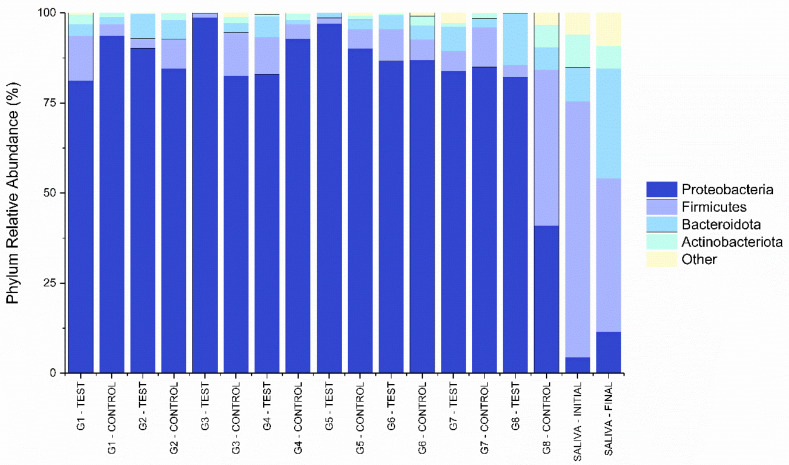
Percentages of most abundant bacterial phyla for the libraries obtained from the different IAI’s.

**Figure 5 materials-16-03904-f005:**
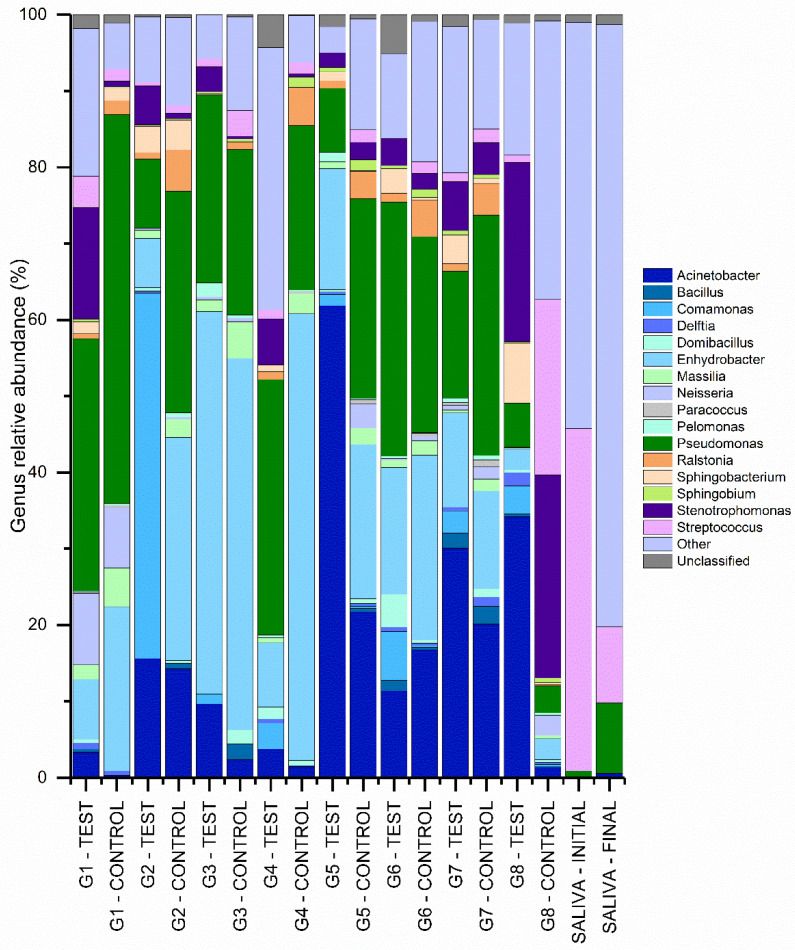
Percentages of most abundant bacterial genera for the libraries obtained from the different IAI’s.

**Figure 6 materials-16-03904-f006:**
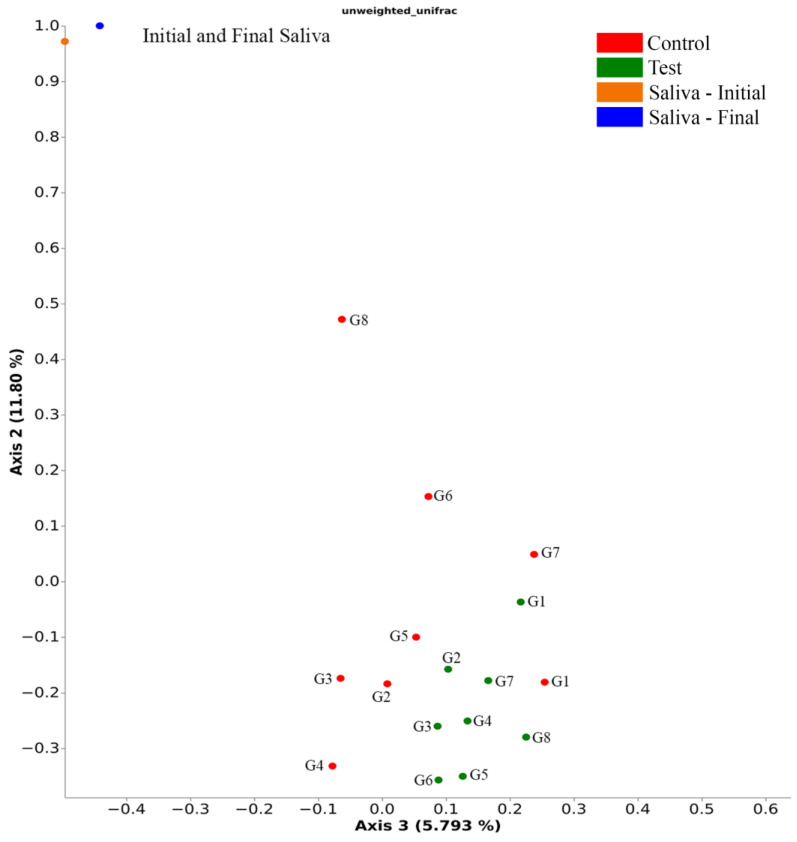
Unweighted Unifrac distances showing test and control groups and the groups individually (PERMANOVA, Pseudo-F = 1.28, number of permutations = 999, *p* = 0.042).

**Figure 7 materials-16-03904-f007:**
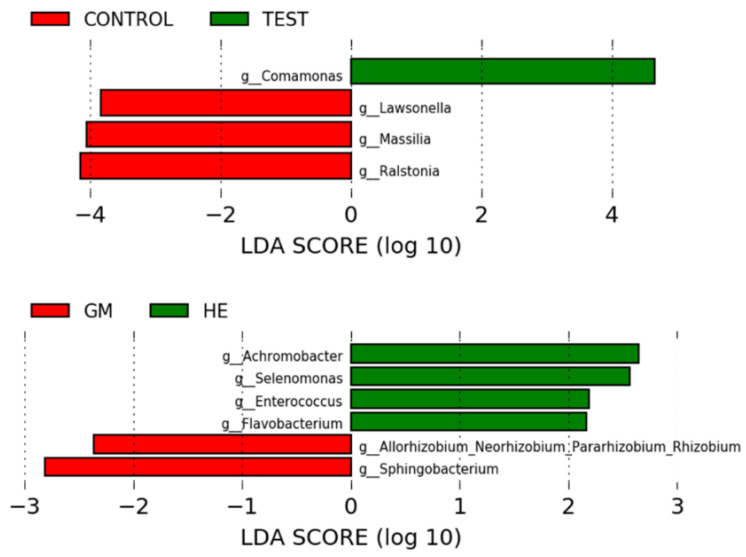
LEfSe results of the comparisons made between platforms and treatment.

**Figure 8 materials-16-03904-f008:**
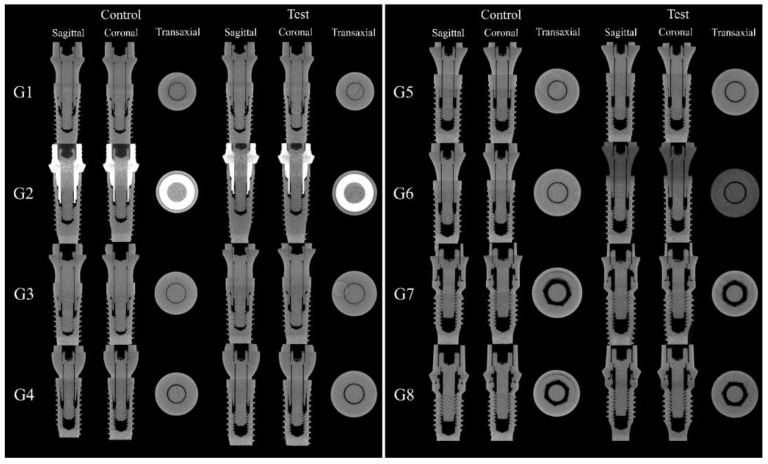
Sagittal, coronal, and transaxial MicroCT sections images in the IAI region of the groups tested.

**Figure 9 materials-16-03904-f009:**
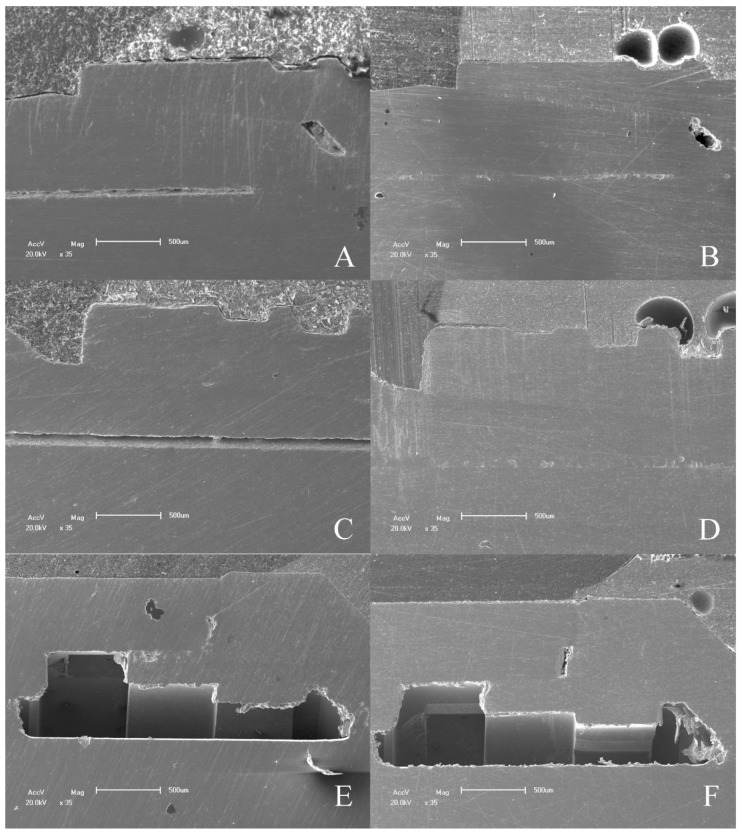
SEM micrographs of IAI tested. (**A**) 16° IAI control; (**B**) 16° IAI tested; (**C**) 11.5° IAI control; (**D**) 11.5° IAI tested; (**E**) EH IAI control; (**F**) EH IAI tested.

**Figure 10 materials-16-03904-f010:**
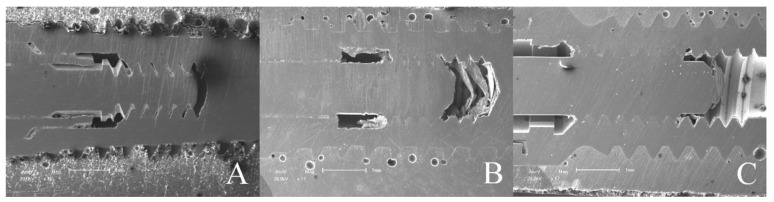
Prosthetic screw threads of prosthetic platforms tested. (**A**) 16° IAI; (**B**) 11.5° IAI; (**C**) EH.

**Figure 11 materials-16-03904-f011:**
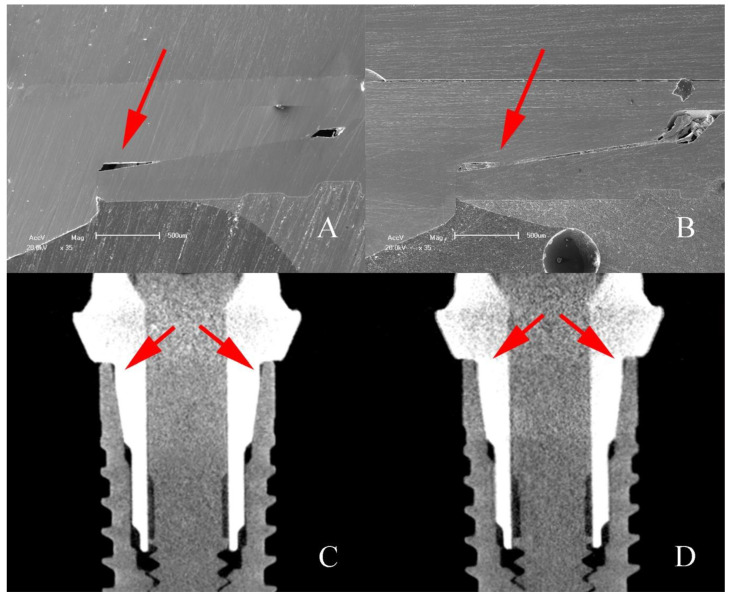
SEM micrographs and MicroCT sagittal sections of G2 implants showing the void space (arrows) found on IAI region. (**A**,**C**) Control; (**B**,**D**) after mechanical loading.

**Table 1 materials-16-03904-t001:** Information about the dental implants, prosthetic platforms, and prosthetic components.

Implants	Connections	Abutments	Torque (Ncm)	Groups n = 15	Brand
Grand Morse Helix (3.5 × 11.5 mm) GM	Internal conical connection (16°)	GM Universal Abutment (4.5 × 6.0 × 3.5 mm)	20	G1	Neodent^®^, Curitiba, Brazil
		GM CoCr Abutment (4.0/4.3 mm)		G2	
		GM Titanium Base (4.5 × 6.0 × 3.5 mm)		G3	
Cone Morse Alvim (3.5 × 11.5 mm) CM	Internal conical connection (11.5°)	2-Piece CM Non-Indexed Universal Abutment (4.5 × 6.0 × 3.5 mm)	15	G4	
		Titanium Base C for CM (3.5 mm)		G5	
		2-Piece CM Indexed Universal Abutment (4.5 × 6.0 × 3.5 mm)		G6	
Titamax Ti EX (3.75 (4.1) × 9.0 mm) HE	External Hexagon	Titanium Base C for EH (4.1/4.3 mm)	32	G7	
		UCLA (4.5 mm)		G8	

**Table 2 materials-16-03904-t002:** Pairwise comparison of statistical analysis.

	G1	G2	G3	G4	G5	G6	G7	G8
G1	-	0.074	0.143	0.993	0.998	0.437	1.000	0.798
G2	0.074	-	1.000	0.008 *	0.012 *	>0.001 *	0.062	0.828
G3	0.143	1.000	-	0.018 *	0.027 *	>0.001 *	0.122	0.936
G4	0.993	0.008 *	0.018 *	-	1.000	0.902	0.996	0.303
G5	0.998	0.012 *	0.027 *	1.000	-	0.846	0.999	0.379
G6	0.437	>0.001 *	>0.001 *	0.902	0.846	-	0.483	0.013 *
G7	1.000	0.062	0.122	0.996	0.999	0.483	-	0.757
G8	0.798	0.828	0.936	0.303	0.379	0.013 *	0.757	-

* Indicates significant difference on comparison (*p* < 0.05).

## Data Availability

The data that support the findings of this study are available from the corresponding author upon reasonable request.

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
