# Peer review of "Bacterial Detection, Deformation, and Torque Loss on Dental Implants with Different Tapered Connections Compared with External Hexagon Connection after Thermomechanical Cycling"

_materials, 2023, doi:10.3390/ma16113904_

Round 1

Reviewer 1 Report

            Dear Authors,

Please find below some recommendations concerning your article entitled” Bacterial detection, deformation and torque loss on dental implants with different tapered connections compared with external hexagon connection after thermomechanical cycling.”

In the Abstract section:

-line 18- SEM, MicroCT- The first occurrence must be described and then the acronym can be used. Please correct it.

Keywords: should match PubMed MeSH terms

In the Materials and Methods section:

Please take into consideration that tables and figures must be included in the text, closest to where they are mentioned:

- Please move Figure 1 after line 95.

- Make a new paragraph” Each evaluated group combines one of the connection designs with a different abutment (Table 1).”

- Please move Table 1 after the that.

- Make a new paragraph” Dental implants (n=120) were used in this study, subdivided into 8 groups (n=15)...”.

In the Results section:

-Please take into consideration that tables and figures must be included in the text, closest to where they are mentioned.

- Please move Table 2 and Figure 2 after line 258

- Table 2 please add a row with the p values and provide a legend about the complete meaning of” a, b, c, bc, abc”

- Please move Figure 3 after line 279

- Please move Figure 4 and 5 after line 286

- Please move Figure 6 after line 314

- Please move Figure 7 after line 316,” A discriminative rate between groups based on relative abundance was obtained by analysis with LEfSe (Figure 7).”

- Please move Figures 8 and 9 after lines 328-330,” Micro CT and SEM analyzes showed that no microgaps were seen in the IAI region and in the contact region between the screw head and the abutment, as can be seen in Figure 8 and Figure 9.”

- Make a new paragraph” The MicroCT and SEM analysis...”

-Please move Figure 10 after lines 333-334,” The groups with 11.5° IAI (CM - G4, G5 and G6) show shorter screws with less thread contact compared to the other platforms tested (Figure 10).”

- Make a new paragraph” Qualitative analysis of the images obtained through on the dental implant used as a control and on the dental implant after...” and insert Figure 11.

Author Response

Authors' responses to REVIEWER 4 comments:

  Dear Authors,

Please find below some recommendations concerning your article entitled” Bacterial detection, deformation and torque loss on dental implants with different tapered connections compared with external hexagon connection after thermomechanical cycling.”

In the Abstract section:

-line 18- SEM, MicroCT- The first occurrence must be described and then the acronym can be used. Please correct it.

Keywords: should match PubMed MeSH terms

Corrected.

In the Materials and Methods section:

Please take into consideration that tables and figures must be included in the text, closest to where they are mentioned:

- Please move Figure 1 after line 95.

- Make a new paragraph” Each evaluated group combines one of the connection designs with a different abutment (Table 1).”

- Please move Table 1 after the that.

- Make a new paragraph” Dental implants (n=120) were used in this study, subdivided into 8 groups (n=15)...”.

All changes have been made.

In the Results section:

-Please take into consideration that tables and figures must be included in the text, closest to where they are mentioned.

- Please move Table 2 and Figure 2 after line 258

- Table 2 please add a row with the p values and provide a legend about the complete meaning of” a, b, c, bc, abc”

- Please move Figure 3 after line 279

- Please move Figure 4 and 5 after line 286

- Please move Figure 6 after line 314

- Please move Figure 7 after line 316,” A discriminative rate between groups based on relative abundance was obtained by analysis with LEfSe (Figure 7).”

- Please move Figures 8 and 9 after lines 328-330,” Micro CT and SEM analyzes showed that no microgaps were seen in the IAI region and in the contact region between the screw head and the abutment, as can be seen in Figure 8 and Figure 9.”

- Make a new paragraph” The MicroCT and SEM analysis...”

-Please move Figure 10 after lines 333-334,” The groups with 11.5° IAI (CM - G4, G5 and G6) show shorter screws with less thread contact compared to the other platforms tested (Figure 10).”

- Make a new paragraph” Qualitative analysis of the images obtained through on the dental implant used as a control and on the dental implant after...” and insert Figure 11.

All changes have been made.

To improve understanding, we changed the positioning of Figure 2 and Table 2, which were corrected as requested.

Reviewer 2 Report

Introduction

- the topic of the article is very actual, since the possible bacterial colonization of the IAI is currently discussed among the scientific community.  To complete the introduction, add a little paragraph describe why an abutment is commonly used to connect the implant to the prostheses in contrast to connect the implant directly to the prostheses. 

- Following the previous points, discuss the possibility to avoid the use of abutments  when using tissue level implants. For this propose, discuss and cite the following article 

Carossa, M.; Alovisi, M.; Crupi, A.; Ambrogio, G.; Pera, F. Full-Arch Rehabilitation Using Trans-Mucosal Tissue-Level Implants with and without Implant-Abutment Units: A Case Report. Dent. J. 2022, 10, 116. https://doi.org/10.3390/dj10070116

Materials and methods

- ''The design of this study (Figure 1) was carried out considering...'' Remove figure 1 from this sentence. Add it in a separate sentence, example: Flowchart of the study is shown in Figure 1.

- What is the rational of performing the study with the crowns made with provisional resin? Since different materials for final crowns, example zirconia or metal-ceramic crowns, have different mechanical properties compared to provisional resin, they can also apply different forces on the IAI and, therefore, the result of the study may not display what can happen with final crown worn in the mouth for years. This point is crucial for me and therefore clarify it and add it as a strong limitation of the study.

Discussion

- Discuss if the result of the study may one reflected also when using short implants. For this porpuse discuss and cite the following article doi: 10.1111/cid.13113

Author Response

Authors' responses to REVIEWER 1 comments:

We tried to revise the entire text to make it more comfortable for readers.

Introduction

- the topic of the article is very actual, since the possible bacterial colonization of the IAI is currently discussed among the scientific community.  To complete the introduction, add a little paragraph describe why an abutment is commonly used to connect the implant to the prostheses in contrast to connect the implant directly to the prostheses.

In contrast to your request, another reviewer requested that the Introduction be shortened, which would be too long. Considering the breadth and importance of the theme, we understand that we could keep the Introduction as it is, but the text was reviewed to better understanding. The approach of yet another alternative design to eliminate the presence of abutments, not included in this study, ended up not being done in order not to lengthen the text even further. Anyway, this has been included in the Discussion, as requested.

- Following the previous points, discuss the possibility to avoid the use of abutments  when using tissue level implants. For this propose, discuss and cite the following article

Carossa, M.; Alovisi, M.; Crupi, A.; Ambrogio, G.; Pera, F. Full-Arch Rehabilitation Using Trans-Mucosal Tissue-Level Implants with and without Implant-Abutment Units: A Case Report. Dent. J. 2022, 10, 116. https://doi.org/10.3390/dj10070116

Provided as requested.

Materials and methods

- ''The design of this study (Figure 1) was carried out considering...'' Remove figure 1 from this sentence. Add it in a separate sentence, example: Flowchart of the study is shown in Figure 1.

Provided as requested. Figure 1 has been changed to respond to another reviewer's suggestion.

- What is the rational of performing the study with the crowns made with provisional resin? Since different materials for final crowns, example zirconia or metal-ceramic crowns, have different mechanical properties compared to provisional resin, they can also apply different forces on the IAI and, therefore, the result of the study may not display what can happen with final crown worn in the mouth for years. This point is crucial for me and therefore clarify it and add it as a strong limitation of the study.

An important issue for this research project was cost and we did some pilots to test the material of the crowns. The 3D printed resins performed well, without any compromise in the test performed. They were also chosen for the ease of obtaining new crowns, if necessary during the test run. Throughout the procedure the crowns were monitored and there was no issue with marginal integrity or any other effect that would cause problems.

The limitation of the study in this aspect was pointed as suggested.

Discussion

- Discuss if the result of the study may one reflected also when using short implants. For this porpuse discuss and cite the following article doi: 10.1111/cid.13113

Provided as requested.

Reviewer 3 Report

Dear Authors, congratulations on your article,

The introduction is well-structured and provides a good overview of the background and research gap. The purpose of the study is clearly stated, and the research question and null hypothesis are well-defined. However, the language could be made more concise and simplified for better readability. The article could benefit from some editing to improve clarity and coherence, especially in the latter half of the introduction. Additionally, it would be helpful to provide more context on the specific internal conical platforms tested and their taper angles to aid readers in understanding the study design and results. Overall, the introduction presents a strong foundation for the study, but some revisions and improvements would enhance its effectiveness.

The Material and Methods section is well-structured, and the information is presented clearly, making it easy to follow. However, there are some minor points that could be improved:

  • - The number of participants in the study is provided (n=120), but it is not clear how many implants were included in each group. This information could be added to the text.
  • - It is not clear why three of the implant groups were subjected only to thermocycling and not mechanical loading. Some justification for this decision would be helpful.
  • - The section on thermomechanical cycling under saliva contamination is detailed and well-explained. However, it could benefit from more information about how the mechanical loadings and thermic cycles were performed during the testing period (e.g., how often they were applied, for how long, etc.).

The discussion section is well written and presents the findings and limitations of the study clearly. It would be helpful to provide a brief explanation of the clinical implications of the findings. This would help readers to understand the practical significance of the results. Finally, it would be useful to suggest avenues for future research based on the limitations of the study.

Minor language revisions required

Author Response

Authors' responses to REVIEWER 2 comments:

We tried to revise the entire text to make it more comfortable for readers.

Dear Authors, congratulations on your article,

The introduction is well-structured and provides a good overview of the background and research gap. The purpose of the study is clearly stated, and the research question and null hypothesis are well-defined. However, the language could be made more concise and simplified for better readability. The article could benefit from some editing to improve clarity and coherence, especially in the latter half of the introduction. Additionally, it would be helpful to provide more context on the specific internal conical platforms tested and their taper angles to aid readers in understanding the study design and results. Overall, the introduction presents a strong foundation for the study, but some revisions and improvements would enhance its effectiveness.

 The Material and Methods section is well-structured, and the information is presented clearly, making it easy to follow. However, there are some minor points that could be improved:

  • - The number of participants in the study is provided (n=120), but it is not clear how many implants were included in each group. This information could be added to the text.

Provided as requested.

  • It is not clear why three of the implant groups were subjected only to thermocycling and not mechanical loading. Some justification for this decision would be helpful.

The main test is this study was the bacterial detection by 16S rRNA sequencing. In a pilot test we noted that three sets were the minimum for this test. Thus, given the limited number of sets available, we reserved 3 for the control group and 10 for the test group.

  • - The section on thermomechanical cycling under saliva contamination is detailed and well-explained. However, it could benefit from more information about how the mechanical loadings and thermic cycles were performed during the testing period (e.g., how often they were applied, for how long, etc.).

Provided as requested.

The discussion section is well written and presents the findings and limitations of the study clearly. It would be helpful to provide a brief explanation of the clinical implications of the findings. This would help readers to understand the practical significance of the results. Finally, it would be useful to suggest avenues for future research based on the limitations of the study.

Provided as requested.

Reviewer 4 Report

The study is interesting and genuine, however the authors should address the following points to improve the quality of the manuscript:

- Please add a short statement in the abstract to emphasize on the current research gap and question.

- The introduction section seems a bit long. Please reduce it to the most relevant information that support the introduction section.

- Although IRB approval number was mentioned at the end of the article, please mention it in the materials and methods section.

- Line 106: please mention the details of "parallelometer" used for implant mounting.

- Line 110: what was the rationale of using 3D printed temporary crowns, why not definitive crown materials? as marginal integrity and thermal cycling may have different effect on the study outcomes.

- Line 118: why did the authors fix the crowns with polyether? why not temporary cement for simulation of real clinical scenarios?

- Line 123: were subjects randomly selected for saliva draw?

- Torque and torque loss section should be expanded for clarity.

- Figure 6 is not clear. Please improve its resolution.

- The discussion is too long. Please reduce its length accordingly.

- Please add the study limitations and directions for future research in the discussion section.

- Please expand the conclusion part and summarize it in bullet section.

Author Response

Authors' responses to REVIEWER comments:

We tried to revise the entire text to make it more comfortable for readers.

The study is interesting and genuine, however the authors should address the following points to improve the quality of the manuscript:

- Please add a short statement in the abstract to emphasize on the current research gap and question.

Provided as requested.

- The introduction section seems a bit long. Please reduce it to the most relevant information that support the introduction section.

We know that the Introduction was a little long, but still other reviewers asked for additional information to be included. Considering the breadth and importance of the topic, we added the requested information, and therefore there was no reduction of this section.

- Although IRB approval number was mentioned at the end of the article, please mention it in the materials and methods section.

The IRB approval number was already in the text, but we corrected it to make it more explicit.

- Line 106: please mention the details of "parallelometer" used for implant mounting.

Provided as requested and the Figures 1 has been changed to display the parallelometer used.

- Line 110: what was the rationale of using 3D printed temporary crowns, why not definitive crown materials? as marginal integrity and thermal cycling may have different effect on the study outcomes.

An important issue for this research project was cost and we did some pilots to test the material of the crowns. The 3D printed resins performed well, without any compromise in the test performed. They were also chosen for the ease of obtaining new crowns, if necessary during the test run. Throughout the procedure the crowns were monitored and there was no issue with marginal integrity or any other effect that would cause problems.

- Line 118: why did the authors fix the crowns with polyether? why not temporary cement for simulation of real clinical scenarios?

In the same way we did for the choice of crowns, we also tested their fixation with polyether. We had no problems, there was no loosening, the material provided adequate fixation and sealing capacity of the coronal part and, importantly, also allowed easier removal for access to the abutments after the proposed tests.

- Line 123: were subjects randomly selected for saliva draw?

Saliva donors were volunteers in order of answering the call and selected only and only when meeting the requirements: did not present oral manifestations of systemic diseases or acute infectious in the oral cavity and availability for the other necessary collections.

- Torque and torque loss section should be expanded for clarity.

Provided as requested.

- Figure 6 is not clear. Please improve its resolution.

Provided as requested.

- The discussion is too long. Please reduce its length accordingly.

As with the Introduction, other reviewers asked that some aspects be added, which was done. We also understand that the breadth and importance of the theme justify that the section be bigger to allow the correct approach.

- Please add the study limitations and directions for future research in the discussion section.

We had already pointed out some limitations, but in response to your request, as well as that of other reviewers, we have rewritten it to better clarify it.

- Please expand the conclusion part and summarize it in bullet section.

Provided as requested.

Round 2

Reviewer 2 Report

The authors revised the manuscript correctly.